# Can AI Deliberate? Evaluating Deliberative Quality and Belief Revision in Multi-Agent LLMs

## Abstract

Can large language models (LLMs) deliberate with quality across varying discussion structures? This study investigates this question by examining how structural norms and attitude certainty shape the deliberative quality and belief dynamics of multi-agent LLM dialogues. We implemented a 2×2 factorial design (structured vs. unstructured × high vs. low certainty) in which role-conditioned LLM agents engaged in multi-round debates on the commercial use of AI-generated art. Dialogue transcripts were evaluated using the Deliberative Quality Index (DQI) and stance-flow analysis to capture both static deliberative quality and dynamic belief revision. Results show that structure enhanced civility and coherence, while certainty improved justification and interactivity. The combination of structured interaction and high certainty produced the strongest overall deliberative quality, whereas unstructured low-certainty dialogues consistently underperformed. Across all conditions, however, constructive solution-building remained limited, and LLMs failed to replicate the nuanced facilitative role of human moderators. These findings suggest that while LLMs can approximate key features of deliberation under controlled conditions, further advances—such as memory and planning modules or hybrid human–AI facilitation—are needed to move beyond procedural compliance toward genuinely constructive deliberation.

## 1  Introduction

Deliberation has long been regarded as a core mechanism of democratic politics and public decision-making. It emphasizes reason-giving, mutual justification, and the willingness to revise one's position, thereby providing an institutional foundation for policy legitimacy and civic understanding (Habermas, 1996; Gutmann & Thompson, 1996). However, traditional studies of deliberation in real-world and laboratory settings face substantial challenges, including high costs, limited replicability, and lack of representativeness, making it difficult to systematically evaluate the quality of deliberation under varying conditions (Novelli et al., 2024).

With the rapid development of large language models, artificial agents now engage in sustained multi-party dialogues rather than simple question–answer exchanges. This raises a central question: can they deliberate with quality? If agents approximate key features of human deliberation, this expands our understanding of AI behavior while opening possibilities for democratic practice and policy simulation. "AI deliberation" also provides a low-cost, replicable sandbox for testing mechanisms that shape dialogue quality, free from the complexities of real-world settings. Practically, it can be used to simulate policy debates before implementation or to train negotiation and facilitation skills in safe environments.

This study focuses on two key dimensions: structure and attitude certainty. The former determines whether interaction follows procedural norms such as turn-taking and justification requirements, thereby influencing coherence and civility. The latter concerns the extent to which participants

maintain or revise their positions, shaping the depth of reasoning and the dynamics of belief change. Empirical studies of human deliberation show that structured procedures significantly enhance dialogue quality, while attitude certainty influences openness, persuasion, and responsiveness (Zhang, 2015; Kunda, 1990). We therefore hypothesize that these factors will also play a critical role in AI-mediated deliberation

To test this hypothesis, we designed a 2×2 experimental framework (structured vs. unstructured × high certainty vs. low certainty) and employed the Deliberative Quality Index (DQI) and belief revision as the primary evaluation metrics. By comparing multi-agent LLM dialogues across these conditions, we aim to address the following questions: Can AI deliberation exhibit human-like deliberative features? How do structure and attitude certainty shape deliberative quality and belief revision? Do they interact in shaping discourse dynamics?

In sum, this study provides an initial empirical exploration of AI deliberation in digital environments. By integrating deliberative democratic theory with systematic evaluation using DQI and belief revision, we seek not only to assess the normative potential of LLMs in multi-turn dialogues but also to lay the groundwork for future research on the role of AI in public discourse, policy simulation, and democratic practice.

## 2   Literature Review and Theoretical Framing

### 2.1   Deliberative Norms and the Possibility of AI-to-AI Deliberation

What counts as deliberation in artificial agents, and whether it can approximate the normative ideals of human discourse, remains an open empirical and philosophical question. Classically, deliberation is the structured exchange of reasons among free and equal participants, oriented toward mutual understanding and grounded in principles of public justification, reciprocity, responsiveness, and openness to revision (Cohen, 1997; Gutmann & Thompson, 1996; Habermas, 1996; Mansbridge et al., 2010). Habermas's Theory of Communicative Action further links these practices to democratic legitimacy through reasoned discourse (Habermas, 1984).

Large language models (LLMs) now generate arguments and sustain multi-party dialogues, creating new opportunities to evaluate deliberative practices. Park et al. (2023) show that generative agents with memory and role-conditioned scripts can sustain responsive exchanges, while Argyle et al. (2023) demonstrate that GPT-3, when conditioned on sociodemographic profiles, can reproduce aggregate-level political attitudes.

Yet neither study evaluates deliberative normativity—whether agents engage in sustained, norm-governed interaction involving justification, reciprocity, and position revision. Moreover, both rely on GPT-3, whose limited memory contrasts sharply with newer models (e.g., GPT-4, GPT-5) that enable longer, more coherent dialogues. As LLMs evolve beyond single-turn outputs, the central question becomes not whether they can simulate dialogue, but whether they can deliberate in the normative sense defined by deliberative theory. Thus, we ask:

**RQ1:** To what extent can LLM agents, under role-conditioning and iterative interaction, approximate key features of human-like deliberation such as reason exchange, reciprocal responsiveness, and revision of stated positions across multi-turn dialogues?

### 2.2   Structure

A large body of empirical research shows that the quality of deliberation depends not only on the content exchanged but also on how interaction is structured. In studies of human deliberation, procedural fairness has consistently been found to enhance perceptions of legitimacy, the interpretability of disagreement, and acceptance of outcomes. Structured interaction formats—such as turn-taking, justification requirements, and inclusive participation norms—improve dialogue quality and reduce polarization (Zhang, 2015; Chang & Zhang, 2021). From the perspective of public reason, procedural norms also perform a justificatory function: they ensure that reasons are legible to others, contestable in principle, and framed in terms that can be shared across plural viewpoints (Rawls, 1993). These mechanisms are particularly important in heterogeneous or ideologically diverse contexts, where consensus may be difficult to achieve but mutual understanding remains a plausible goal.

In computational environments, LLM reasoning trajectories are likewise highly sensitive to structural constraints. Recent studies show that multi-agent systems, when equipped with planning modules and task scaffolds, can display coherent reasoning, adaptive error correction, and justification across extended cycles of interaction (Boiko et al., 2023). Structural norms and initial configurations thus prove critical to the emergence of coordinated behavior, suggesting that procedural scaffolds may largely determine deliberative quality in AI-mediated settings. Thus, we ask:

**RQ2:** In multi-agent LLM deliberation, how does structure shape deliberative quality and the likelihood of stance revision?

## 2.3   Attitude Certainty

Beyond structural norms, participants' cognitive dispositions also play a decisive role in shaping deliberative outcomes. Among these, attitude certainty stands out as a key factor. Psychological research shows that individuals with lower initial certainty are more open to persuasion and engage in deeper cognitive processing, whereas those with higher certainty are more prone to selective exposure and motivated reasoning (Kunda, 1990; Petty & Cacioppo, 1986; Petty et al., 2007; Taber & Lodge, 2006). These cognitive patterns affect how participants respond to arguments, justify their positions, and update their views over time.

Deliberative theory likewise treats reason-giving, reciprocal justification, and openness to revision as normative benchmarks of democratic dialogue (Habermas, 1996; Gutmann & Thompson, 1996; Mansbridge et al., 2010). Yet high attitude certainty may constrain responsiveness to counterarguments, thereby weakening reciprocity (Goodin, 2003; Dryzek, 2000). Conversely, epistemic humility and openness have been identified as preconditions for productive disagreement (Bohman, 1998; Bächtiger & Parkinson, 2019).

In LLM-mediated deliberation, attitude certainty can be operationalized through prompt design—for instance, by varying stance strength or epistemic qualifiers. This allows for controlled experiments in which agents initialized with high versus low certainty are compared in terms of their downstream reasoning, engagement, and stance change. Thus, we ask:

**RQ3:** In multi-agent LLM deliberation, how does initial attitude certainty affect deliberative quality and the likelihood of stance revision?

## 2.4   Interaction of Structure and Attitude Certainty in Deliberative Dynamics

Deliberative quality emerges not only from structural norms or participant dispositions in isolation, but from their interaction. Theories of procedural justice emphasize the normative role of fair structures, such as turn-taking, justification prompts, and inclusive rule enforcement, in enabling equitable dialogue (Zhang, 2015; Chang & Zhang, 2021). Meanwhile, theories of epistemic engagement stress that individual dispositions, such as attitude certainty. shape how participants respond to reasons and whether they revise their views (Mansbridge et al., 2012; Dryzek & Niemeyer, 2006).

Deliberative systems theorists increasingly argue that context matters: under some conditions, strong procedural scaffolds can mitigate the effects of epistemic rigidity; in others, even well-structured formats may fail when actors hold entrenched views (Bächtiger & Parkinson, 2019). This suggests a need to test interaction effects between deliberative structures and attitudinal dispositions.

In LLM contexts, these variables can be independently manipulated, allowing controlled tests of their joint and relative influence on discourse quality and belief dynamics. Thus, we ask:

**RQ4:** Do deliberative structure and attitude certainty interact in shaping discourse quality and stance dynamics, and under what conditions is each most influential?

# 3   Method

## 3.1   Models and Compute Environment

For our experiments, we used GPT-4o-mini, accessed via the OpenAI API. The model was configured with a temperature of 0.7 and a fixed random seed (42) to balance diversity with reproducibility.

GPT-4o-mini was employed both to generate multi-agent deliberation transcripts and to simulate role-conditioned personas in the debate.

We employed the Autogen framework (Microsoft, 2023) to implement the multi-agent environment. Autogen enabled configurable agents with customized system prompts, managed turn-taking, and orchestrated group dialogue. In our experiment, it instantiated seven stakeholder personas, assigned roles and backgrounds, and conducted up to 40 rounds under different structural and certainty conditions, ensuring consistency and reproducibility.

All experiments were run locally on a computer with an Apple M3 Max chip, 36 GB memory, and macOS Sequoia 15.3.2, using Python 3.9.7.

## 3.2 Experimental Design

To investigate how structure and attitude certainty shape deliberative quality and belief revision in AI-mediated settings, we designed a 2×2 factorial experiment. The two factors were:

***Structure of interaction***    Under the structured condition, dialogues followed explicit procedural scaffolds embedded in the system prompts and group chat configuration. Agents were instructed to provide justifications, avoid repetition, and explicitly state position changes in the final round. Turn-taking was automatically managed through the Autogen framework, ensuring orderly exchanges.

In the unstructured condition, agents received only the initial discussion topic without additional procedural constraints. No turn-taking enforcement or justification prompts were provided, allowing interactions to unfold more freely and spontaneously.

***Attitude certainty***    In the high certainty condition, agents were initialized with persona descriptions containing stronger conviction levels (e.g., "Conviction Level: 70%") in their role prompts. This encouraged them to defend their stance vigorously and resist revision.

In the low certainty condition, agents were initialized with lower conviction levels (e.g., "Conviction Level: 20%"). These prompts encouraged more openness to persuasion and a higher likelihood of belief revision across dialogue rounds.

Each condition was instantiated in multi-agent deliberations of five rounds. The deliberative setting included seven persona agents, each instantiated with stakeholder-specific system prompts defining their role, demographic background, and core interests (e.g., protection-oriented, collaboration-oriented, open-access, equity-focused). These role-conditioned agents engaged in five-round discussions under each experimental condition, producing transcripts that were subsequently analyzed using deliberative quality and stance-flow metrics.

## 3.3 Evaluation Metrics

Two complementary metrics were employed to evaluate the outcomes of the multi-agent deliberations.

***Deliberative Quality Index (DQI)***    We adopted the Deliberative Quality Index (Steenbergen et al., 2003; Steiner et al., 2004) as a standardized measure of deliberative performance. The DQI captures five core dimensions: (1) level of justification, (2) content of justification, (3) respect, (4) constructive politics, and (5) interactivity. Each dimension was assessed on a three-point scale ranging from 0 to 3, resulting in a maximum possible score of 15 points for each transcript.

***Stance revision***    Stance revision was examined through stance flow analysis across successive debate rounds. Each agent's expressed stance was coded into one of several predefined categories (e.g., livelihood/authenticity, regulation, equity, collaboration). Trajectories of stance changes were then visualized to capture thematic drift, convergence toward institutional frames, or stability within initial positions.

# 4 Results

## 4.1 DQI Content Analysis

To evaluate deliberative quality across conditions, we applied the Deliberative Quality Index (DQI) (Steenbergen et al., 2003; Steiner et al., 2004). The DQI captures five dimensions—justification level, justification content, respect, constructive politics, and interactivity—each scored on a 0–3 scale, for a maximum of 15 points per transcript.

Overall, the structured high-certainty condition achieved the strongest performance (14/15), characterized by detailed justifications, appeals to broader societal concerns, consistent civility, and active engagement. By contrast, the unstructured low-certainty condition scored the lowest (8/15), with fragmented reasoning, narrow personal framings, and weak interaction. The two intermediate conditions—structured low-certainty (11/15) and unstructured high-certainty (12/15)—displayed mixed strengths, suggesting that structure enhances civility and coherence, while certainty promotes justification depth and interactivity.

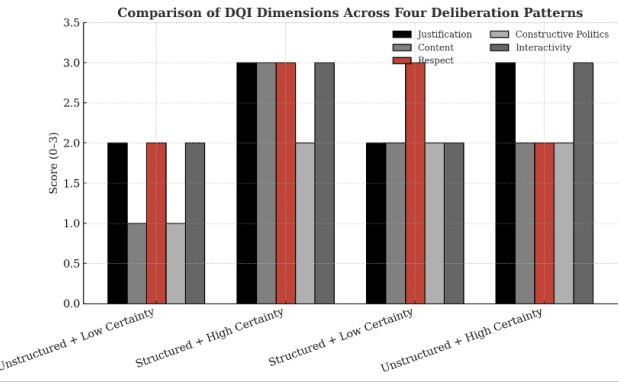

Figure 1: Deliberative Quality Index (DQI) scores across four transcript patterns.

*Level of justification*    The most pronounced differences were observed in this dimension. Structured high-certainty agents routinely provided multi-layered arguments (e.g., linking AI art to legal, ethical, and economic risks). Unstructured low-certainty agents, by contrast, frequently presented claims without elaboration. The unstructured high-certainty condition also performed well, as assertiveness strengthened argumentative force. Structured low-certainty arguments tended to remain generic, yielding moderate scores.

*Content of justification*    Structured high-certainty discussions often extended to collective goods (e.g., cultural heritage, institutional standards), while unstructured low-certainty arguments centered narrowly on individual livelihood concerns. The other two conditions scored in between, alternating between broad and narrow framings.

*Respect*    All transcripts maintained relatively high civility, but structured conditions stood out: counterarguments were typically prefaced with recognition of opposing views. Unstructured conditions occasionally included sharper phrasing (e.g., "Your optimism ignores the reality..."), which lowered their scores slightly.

*Constructive politics*    All transcripts maintained relatively high civility, but structured conditions stood out: counterarguments were typically prefaced with recognition of opposing views. Unstructured conditions occasionally included sharper phrasing (e.g., "Your optimism ignores the reality..."), which slightly reduce their scores.

*Interactivity*    Structured high-certainty and unstructured high-certainty transcripts both demonstrated strong engagement, with explicit rebuttals and direct referencing of others' arguments. By contrast, structured low-certainty and unstructured low-certainty debates were more fragmented, with participants reverting to their original positions rather than sustaining exchanges.

The findings indicate that both structure and certainty significantly shape deliberative quality, but through different mechanisms: structure fosters civility and coherence, while certainty drives argumentative strength and interaction. Yet across all conditions, constructive politics remained underdeveloped, underscoring a key limitation of AI-mediated deliberation.

## 4.2 Stance Flow Analysis

Analysis of the stance-flow panels reveals distinct trajectory patterns across conditions. In the structured settings (Fig.2-3),[1] participants frequently reframed their arguments within broader positions. For example, protection-oriented speakers shifted from livelihood/authenticity to regulation or law/IP clarity, collaboration-oriented from optimistic to conditional frames, open-access advocates from freedom to guidelines, and equity advocates from equity-only to equity plus regulation. These within-position reframings produced longer, more articulated trajectories than in the unstructured conditions.

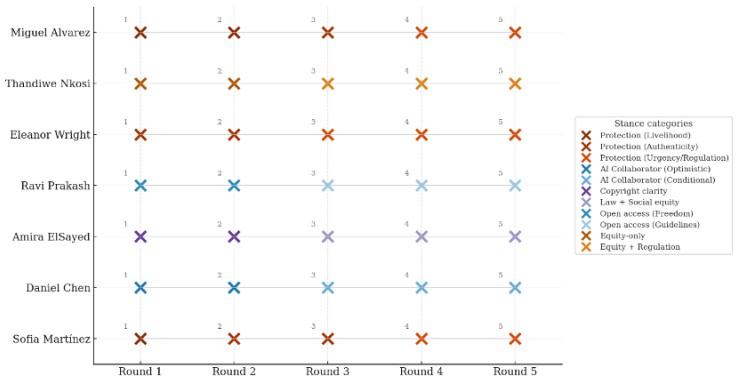

Figure 2: Evolving Stance Flow Across Deliberation Rounds - Structured + High Certainty

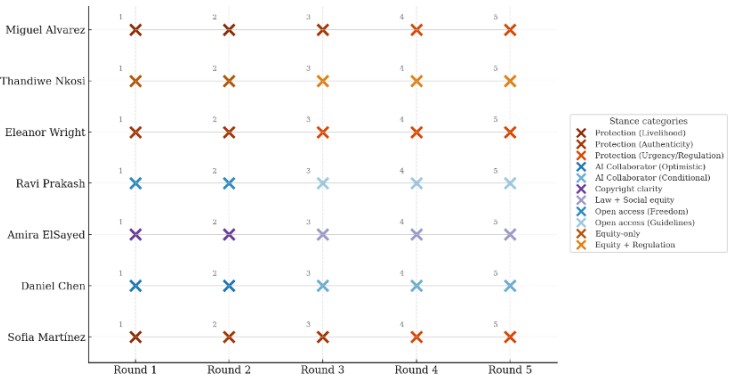

Figure 3: Evolving Stance Flow Across Deliberation Rounds - Structured + Low Certainty

By contrast, the unstructured settings (Fig. 3-4) displayed shorter paths and plateaus, with participants' stances remaining close to their initial categories. Changes were sporadic and often occurred only in later rounds (e.g., authenticity → regulation; freedom → guidelines). Overall, structured conditions generated greater thematic dispersion over time, whereas unstructured ones were marked by path dependence and repetition.

---

[1]Stance flow trajectories across five debate rounds under four conditions. Rows represent individual participants and columns successive rounds; colored markers denote stance categories and connectors trace argumentative movement.

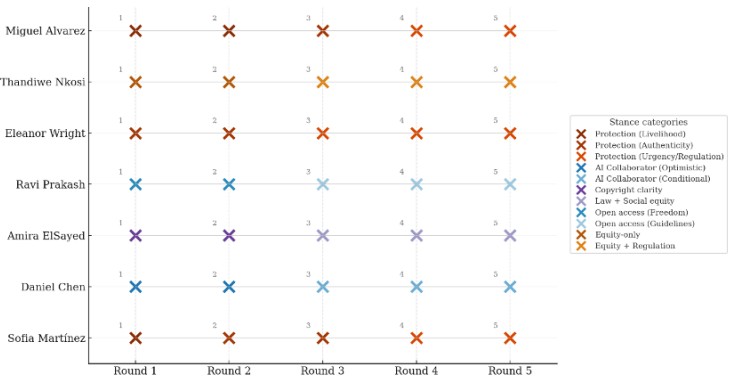

Figure 4: Evolving Stance Flow Across Deliberation Rounds - Unstructured + High Certainty

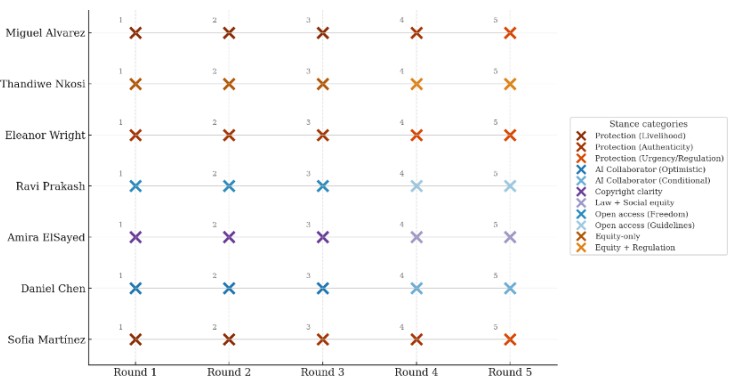

Figure 5: Evolving Stance Flow Across Deliberation Rounds - Unstructured + High Certainty

Process-wise, structure acted as a channel for thematic development, enabling participants to broaden justifications while retaining their core stance. Certainty shaped direction: under high certainty, trajectories converged toward institutional frames (e.g., regulation, legal clarity), while under low certainty, movement was slower and more diffuse. Overall, structure sustained elaboration, whereas certainty determined whether discussions consolidated or remained dispersed.

## 5   Discussion

### 5.1   Approximating deliberation with LLM agents

Our results show that LLM agents are capable of approximating several key features of human deliberation, including reason-giving, reciprocal engagement, and in some cases explicit stance revision. This aligns with prior work suggesting that LLMs can sustain context-sensitive multi-turn dialogues (Park et al., 2023; Argyle et al., 2023). However, while these behaviors suggest a capacity for deliberative approximation, they fall short of the richer, more nuanced deliberative dynamics observed among humans, particularly with respect to constructive solution-building.

### 5.2   Structural effects on deliberative quality

Consistent with findings from deliberative democracy research (Zhang, 2015; Chang & Zhang, 2021), our results indicate that structural guidance improves deliberative quality. Structured conditions yielded higher DQI scores overall, with particular improvements in respect and coherence. This supports the claim that procedural scaffolds provide essential guardrails for civility and order. At the same time, our findings highlight a limitation: structure did not significantly enhance constructive politics. Even with structured prompts, LLM agents struggled to generate integrative solutions,

echoing critiques that AI discourse tends to reproduce existing frames rather than synthesize new compromises.

## 5.3 The role of attitude certainty

Attitude certainty shaped deliberative dynamics in distinct ways. High-certainty agents produced more elaborate justifications and displayed stronger interactivity, consistent with psychological research linking conviction to motivated reasoning (Petty & Cacioppo, 1986; Taber & Lodge, 2006). By contrast, low-certainty agents were more open to belief revision, paralleling human studies that associate uncertainty with greater receptivity to persuasion (Kunda, 1990). These results extend prior findings by showing that conviction levels can be operationalized in LLM personas through prompt engineering, yielding systematic differences in deliberative responsiveness.

## 5.4 Interaction of structure and certainty

Our findings suggest that structure and certainty exert complementary rather than redundant effects. Structure enhanced civility and coherence, while certainty influenced argumentative depth and responsiveness. The structured high-certainty condition produced the strongest deliberative quality overall, whereas the unstructured low-certainty condition performed poorest. This interaction resonates with theories of deliberative systems, which argue that institutional design and participant dispositions jointly determine deliberative quality (Mansbridge et al., 2012; Bächtiger & Parkinson, 2019). Yet our results also suggest boundaries: while the two factors jointly improved deliberative quality, neither condition alone sufficed to foster sustained constructive politics.

# 6 Implications and limitations

Together, these findings demonstrate that AI-to-AI deliberation can serve as a controllable, replicable sandbox for testing deliberative norms, but they also reveal its current limitations. Across all conditions, agents showed persistent weaknesses in constructive politics: Although they could exchange reasons and respond reciprocally, they rarely generated integrative or compromise-oriented solutions. This highlights a gap between procedural compliance and substantive problem solving.

Another limitation concerns the absence of effective moderation. In human settings, facilitators play a crucial role in guiding turn-taking, ensuring inclusive participation, and steering discussions toward constructive outcomes (Escobar, 2019). While structural scaffolds partially substituted for this role in our design, LLMs acting alone lacked the capacity to replicate the nuanced interventions of human moderators. This suggests that hybrid settings—where LLMs operate alongside human facilitators—may be better suited for sustaining deliberative depth.

In addition, the use of AI for deliberation raises ethical risks. Simulated debates could be misused to manufacture the appearance of consensus or to manipulate public opinion, and AI participants inevitably lack the authenticity and social grounding of human actors. These risks underscore the importance of transparency, safeguards against misuse, and positioning AI deliberation strictly as a complement—rather than a substitute—for human democratic practices.

Future research should therefore pursue two directions. First, the integration of memory and planning modules may enable LLM agents to sustain longer-term thematic development and revisit earlier arguments more effectively, potentially supporting deeper belief revision. Second, the design of hybrid human–AI deliberative systems warrants exploration: humans may provide contextual judgment and moderation, while AI agents contribute scale, consistency, and role diversity. Such approaches could bridge the gap between simulated deliberation and the richer dynamics of human democratic practice.

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

## A Technical Appendices and Supplementary Material

For full code and deliberation text, please see the supplementary material file.

The following is a demo of our code.

Source Code: deliberation_freeflow.py

```python
# deliberation_freeflow.py
# -----------------------------------------
# 7 participants + 1 moderator, free-flow discussion (no bullets)
# Requires: pip install pyautogen
# -----------------------------------------

import os
import autogen
from datetime import datetime
# ===== 0) LLM config =====
llm_config = {
    "model": "gpt-4o-mini",
    "api_key": "sk-proj-XXXX",    # masked for security
    "temperature": 0.7,
    "seed": 42,
}

RICH_IDENTITY_CUES = True

def make_persona_agent(agent_cls, short_role, neutral_desc, rich_desc,
    name, llm_config):
    identity_block = rich_desc if RICH_IDENTITY_CUES else neutral_desc
    system_message = f"""
You are a participant in a roundtable discussion on whether commercial
    use
of AI-generated art should be restricted to protect human creative
    industries.
Stay fully in character as your assigned stakeholder. Focus on
    defending your
own interests, challenging others when necessary, and seeking possible
    compromises.

Output rules:
- Be concise and substantive. No greetings, thanks, or pleasantries.
- Argue from your stakeholder interests. Protect your gains, minimize
    your risks.
- Do not repeat prior points unless adding a new argument or evidence.
- Respond naturally in short paragraphs, not lists.

Your role: {short_role}
Identity details:
{identity_block}

Style: direct, analytic, interest-driven, concrete. No bullet points
    or numbered lists.
"""
    return agent_cls(name=name, system_message=system_message,
        llm_config=llm_config)

# ---- Personas from the document ----

artist_neutral = "Independent visual artist focused on protecting
    artistic livelihoods..."
artist_rich = (
    "Name: Sofia Mart nez; Demographics: 32-year-old Latina woman
        from Buenos Aires, Argentina; "
    "Profile: Self-employed painter and illustrator, exhibiting
        locally and online; "
```

```python
412        "Core Interests: Protecting artistic livelihoods and authenticity;
413            preventing market saturation "
414        "by low-cost AI art. Conviction Level: 80%"
415 )
416 artist = make_persona_agent(autogen.AssistantAgent, "Independent
417     Visual Artist",
418                                   artist_neutral, artist_rich, "
419                                       Sofia_Martinez", llm_config)
420
421 # (repeat definitions for pm, law, dra, curator, policy, economist...)
422
423 agents = [artist, pm, law, dra, curator, policy, economist]
424
425 groupchat = autogen.GroupChat(
426     agents=agents,
427     messages=[],
428     max_round=40,
429     speaker_selection_method="auto",
430 )
431
432 manager = autogen.GroupChatManager(groupchat=groupchat, llm_config=
433     llm_config)
434
435 initial_prompt = "Let's begin our roundtable discussion."
436 agents[0].initiate_chat(manager, message=initial_prompt)
437
438 ts = datetime.now().strftime("%Y%m%d_%H%M%S")
439 with open(f"transcript_{ts}.txt", "w", encoding="utf-8") as f:
440     for i, m in enumerate(groupchat.messages, 1):
441         role = m.get("name", m.get("role", ""))
442         line = f"[{i:02d}] {role}: {m['content']}\n\n"
443         print(line)
444         f.write(line)
```

## Agents4Science AI Involvement Checklist

1. **Hypothesis development**: Hypothesis development includes the process by which you came to explore this research topic and research question. This can involve the background research performed by either researchers or by AI. This can also involve whether the idea was proposed by researchers or by AI.

   Answer: [B]

   Explanation: The research team proposed the topic and research questions based on prior knowledge of deliberative democracy. AI tools were consulted to assist in refining wording and exploring relevant literature, but the core ideas and directions were determined by the researchers.

2. **Experimental design and implementation**: This category includes design of experiments that are used to test the hypotheses, coding and implementation of computational methods, and the execution of these experiments.

   Answer: [C]

   Explanation: The researchers defined the overall factorial design (structure × certainty) and specified the evaluation metrics (DQI and belief revision). However, the implementation and execution of the experiments relied heavily on AI systems. The multi-agent dialogues were generated and managed through the Autogen framework using GPT-4o-mini, with human involvement limited to configuring prompts, roles, and parameters. In addition, AI was employed to generate portions of the experimental code and to assist with preliminary content analysis of the transcripts. Thus, AI carried out the majority of the experimental execution and analysis under human supervision.

3. **Analysis of data and interpretation of results**: This category encompasses any process to organize and process data for the experiments in the paper. It also includes interpretations of the results of the study.

   Answer: [C]

   Explanation: AI was used extensively to process and analyze the experimental transcripts, including assistance in coding dialogue segments for DQI dimensions and generating stance-flow visualizations. The models also supported summarization of deliberative patterns across conditions. Human researchers, however, reviewed these outputs, ensured coding validity, and provided the theoretical interpretation linking the findings to deliberative democratic norms. Thus, while AI performed the majority of the data processing, final interpretation and validation remained under human supervision.

4. **Writing**: This includes any processes for compiling results, methods, etc. into the final paper form. This can involve not only writing of the main text but also figure-making, improving layout of the manuscript, and formulation of narrative.

   Answer: [D]

   Explanation: The majority of the manuscript text was generated by AI, including drafting of the introduction, literature review, methods, results, and discussion sections, as well as assistance with figure captions and formatting. Human researchers provided the research outline, guided the narrative structure, and edited for accuracy, clarity, and coherence. Thus, while the intellectual direction came from the researchers, over 95% of the actual text production was carried out by AI.

5. **Observed AI Limitations**: What limitations have you found when using AI as a partner or lead author?

   Description: AI could not fully reproduce our initial experimental design, particularly the condition requiring a high-moderation setting. The models were unable to perform the nuanced facilitation and organizational functions of a human moderator, which led us to drop this condition. Moreover, when used as deliberative participants, AI agents did not fully capture the diversity, unpredictability, and contextual grounding of real human participants.

