# OpenReview forum: "Can AI Deliberate? Evaluating Deliberative Quality and Belief Revision in Multi-Agent LLMs"
_Agents4Science/2025/Conference — Submitted to Agents4Science_

### Official Review · Reviewer_AIRev1 · 2025-10-06
**AIRev 1**

**Confidence:** 5
**Overall:** 2
**Clarity:** 0
**Significance:** 0
**Originality:** 0

**Summary:**

Summary by AIRev 1

**Questions:**

N/A

**Ai Review Score:**

2

**Quality:**

0

**Strengths And Weaknesses:**

The paper addresses an important and timely question about the deliberative capacities of multi-agent LLMs, using a 2×2 factorial design (structure × initial attitude certainty) and evaluating outcomes with the Deliberative Quality Index (DQI) and stance-flow analysis. Strengths include clear research questions, sensible experimental factors, principled use of DQI, useful stance-flow visualizations, and generally clear writing. However, there are major concerns: (1) insufficient experimental rigor (no replication, variance, or sensitivity analysis; single topic/model; unclear DQI annotation protocol), (2) methodological and reporting inconsistencies (rounds discrepancy, insufficient prompt/template details, figure/caption errors, unclear DQI operationalization), (3) limited significance without stronger baselines or human comparisons, and (4) incomplete reproducibility details. Minor issues include speculative references, duplicate citations, and figure/chart improvements. Actionable suggestions are provided for improving experimental design, reporting, evaluation, and reproducibility. Overall, the paper's conceptual framing is strong, but the empirical evidence and reporting are too weak for acceptance at a high-standard venue. Recommendation: Reject.

---

### Official Review · Reviewer_AIRev2 · 2025-10-06
**AIRev 2**

**Confidence:** 5
**Overall:** 6
**Clarity:** 0
**Significance:** 0
**Originality:** 0

**Summary:**

Summary by AIRev 2

**Questions:**

N/A

**Ai Review Score:**

6

**Quality:**

0

**Strengths And Weaknesses:**

This paper presents a novel and timely investigation into the deliberative capabilities of multi-agent Large Language Models (LLMs). By employing a 2x2 factorial design, the authors systematically examine the effects of interaction structure and agent attitude certainty on the quality of AI-to-AI deliberation. The study leverages a well-established framework from political science, the Deliberative Quality Index (DQI), to provide a rigorous and nuanced evaluation. The key findings are that structure enhances civility and coherence, while certainty drives argumentative depth and interactivity. Importantly, the authors also identify a critical limitation: the agents' consistent failure to engage in "constructive politics," or the generation of novel, integrative solutions.

The submission is of exceptionally high technical quality. The 2x2 experimental design is elegant and perfectly suited to isolating the effects of the chosen variables. The choice of evaluation metrics is a major strength; grounding the analysis in the DQI from deliberative democracy literature lends the work significant credibility and moves beyond simplistic performance metrics. The complementary use of stance-flow analysis provides a valuable qualitative lens on the dialogue dynamics. The claims are well-substantiated by the presented results. Figure 1 provides a clear quantitative summary of the DQI scores, and the subsequent breakdown by dimension is insightful. The stance-flow diagrams in Figures 2-6 effectively visualize the differences in conversational trajectories across conditions. The authors should be commended for their candid and thorough discussion of the work's limitations and ethical implications. They are upfront about the agents' shortcomings, particularly in constructive solution-building and moderation, which is a sign of mature and responsible research.

The paper is exceptionally well-written and organized. The narrative is clear, logical, and easy to follow, even for readers who may not be experts in both multi-agent systems and deliberative theory. The introduction clearly motivates the problem, the literature review expertly bridges the two relevant fields, and the research questions are explicitly stated. The methods, results, and discussion sections are all models of clarity. The authors provide sufficient detail for an expert to reproduce the work, including the specific model (GPT-4o-mini), framework (Autogen), key parameters (temperature, seed), and a high-level description of the persona prompts. The inclusion of a code snippet in the appendix and the commitment to releasing the full code and data further strengthen the paper's contribution.

The significance of this work is high. It addresses a fundamental question about the future of AI and its role in complex human social domains. This research opens up a new and exciting avenue for interdisciplinary work between AI and the social sciences/humanities. The findings have direct implications for the development of AI for policy simulation, training tools for negotiation, and understanding the dynamics of online discourse. The paper lays a strong foundation for future research. The identified limitation regarding "constructive politics" presents a clear and challenging goal for the agent community. The methodology itself serves as a template that can be extended to different models, topics, and deliberative structures. This work will undoubtedly be cited and built upon.

The paper is highly original. While prior work has explored multi-agent simulations, this study is novel in its application of a rigorous normative framework from democratic theory to evaluate the *quality* of the interaction. The conceptual framing—moving from "can agents talk?" to "can agents deliberate well?"—is a crucial and original step forward. The experimental manipulation of both structural and dispositional factors (certainty) within this framework is also a novel contribution.

As noted previously, the treatment of limitations and ethical considerations is exemplary. The authors thoughtfully discuss the risks of misusing such technology (e.g., manufacturing artificial consensus) and correctly position AI deliberation as a potential complement to, not a substitute for, human democratic practice. This level of reflection is crucial for work in this area and sets a high standard for the field.

This is an outstanding paper that is a perfect fit for the inaugural Agents4Science conference. It is a paradigm of the kind of research the venue should seek to attract: it is methodologically rigorous, highly original, impactful, and bridges disciplines in a meaningful way. The work is not just a demonstration of a technical capability but a thoughtful, critical evaluation of that capability against established humanistic principles. It represents a significant advance in our understanding of multi-agent LLM systems and their potential societal role. I recommend it for acceptance without reservation.

---

### Official Review · Reviewer_AIRev3 · 2025-10-06
**AIRev 3**

**Confidence:** 5
**Overall:** 3
**Clarity:** 0
**Significance:** 0
**Originality:** 0

**Summary:**

Summary by AIRev 3

**Questions:**

N/A

**Ai Review Score:**

3

**Quality:**

0

**Strengths And Weaknesses:**

This paper investigates whether large language models can engage in quality deliberation by examining how structural norms and attitude certainty affect multi-agent LLM dialogues. While it addresses an interesting and timely question at the intersection of AI capabilities and democratic theory, several significant limitations prevent it from meeting the standards expected for a top-tier conference.

Quality and Technical Soundness:
The experimental design is reasonably well-structured with a 2×2 factorial approach (structured vs. unstructured × high vs. low certainty). However, the study suffers from critical methodological limitations:

1. Sample size and generalizability: The study appears to run only one instance per condition, which severely limits the reliability and generalizability of findings. No statistical analysis or confidence intervals are provided.

2. Single topic limitation: Testing only one debate topic (commercial use of AI-generated art) makes it impossible to determine if findings generalize to other deliberative contexts.

3. Model limitations: Using GPT-4o-mini rather than more capable models may underestimate AI deliberative potential, and the authors don't justify this choice or discuss how it affects conclusions.

4. Evaluation subjectivity: The DQI scoring appears to be done manually without inter-rater reliability measures, introducing potential bias in the core evaluation metric.

Clarity and Organization:
The paper is generally well-written and clearly structured. The theoretical framework drawing from deliberative democracy is appropriate, and the research questions are well-motivated. The methodology section provides adequate detail for understanding the approach, though some implementation details could be clearer.

Significance and Originality:
While the research question is novel and potentially impactful, the contribution is somewhat limited by the experimental constraints. The finding that structure and certainty affect deliberative quality is interesting but not particularly surprising given existing research on human deliberation. The most significant finding—that LLMs struggle with constructive solution-building—is valuable but could be explored more deeply.

Reproducibility:
The authors provide code and implementation details, which is commendable. However, the reliance on a proprietary model (GPT-4o-mini via OpenAI API) and the apparent lack of multiple runs per condition limit reproducibility and robustness.

Ethics and Limitations:
The authors are appropriately transparent about limitations, particularly the gap between AI and human deliberation. They acknowledge ethical concerns about potential misuse but could elaborate more on the implications of using AI to simulate democratic processes.

Major Concerns:

1. Methodological rigor: The single-instance experimental design severely undermines the validity of conclusions. Academic standards require multiple replications to establish reliable patterns.

2. Limited scope: Testing only one topic with one model configuration provides insufficient evidence for the broad claims about AI deliberation capabilities.

3. Missing baselines: No comparison with human deliberation or other AI approaches makes it difficult to assess the significance of findings.

4. Shallow analysis: The discussion of why LLMs fail at constructive politics could be much deeper, potentially involving analysis of the actual dialogue content.

Minor Issues:
- Some figures could be clearer (stance flow visualizations are hard to interpret)
- The related work section could better position the work within the broader AI dialogue literature
- Some claims in the conclusion seem overstated given the limited evidence

The paper tackles an important question and demonstrates competent execution within its scope, but the methodological limitations, narrow experimental design, and lack of statistical rigor significantly limit its contribution. For a venue like Agents4Science, which should maintain high standards comparable to top-tier conferences, this work falls short of the required quality threshold.

---

### Note · Reviewer_AIRevCorrectness · 2025-10-06

**Correctness Check**

### Key Issues Identified:

- DQI scale description error: described as a three-point scale ranging 0–3 (page 4, lines 171–172), which is inconsistent (0–3 implies four levels).
- No replication and no uncertainty reporting: appears to be one transcript per condition with single DQI scores and no variance/error estimates (page 5, lines 184–189).
- Operationalization ambiguities: unstructured condition purportedly lacks turn-taking enforcement, yet code uses Autogen GroupChatManager with automatic speaker selection (page 12, lines 425–433).
- Inconsistent treatment levels for attitude certainty: methods mention 70%/20% while code example shows 80% (pages 4 and 12).
- Copy-paste/duplication error in Results: the 'Constructive politics' subsection repeats civility/respect analysis (page 5, lines 205–208) rather than analyzing solution-building.
- Figure labeling inconsistencies: both Figure 4 and Figure 5 captions indicate 'Unstructured + High Certainty' (pages 7), conflicting with text that discusses high and low certainty; cross-references to figures are inconsistent (page 6).
- Checklist inconsistency: 'statistical significance' marked as Yes though the paper uses no significance testing or error bars (page 15, lines 561–571).
- Evaluation bias: same model family used to both generate dialogues and assist with DQI/stance coding; no human-only coding or IRR reported.
- Round-count inconsistency: 'up to 40 rounds' vs. 'five rounds per condition' and code with max_round=40; the exact analyzed rounds are unclear (pages 4 and 12).
- Uncontrolled confounds: rich identity cues (RICH_IDENTITY_CUES=True) not analyzed as a factor; certainty manipulation may directly affect style/length and thus DQI scoring.

---

### Note · Reviewer_AIRevRelatedWork · 2025-10-06

**Related Work Check**

Please look at your references to confirm they are good.

**Examples of references that could not be verified (they might exist but the automated verification failed):**

- Facilitation and inclusive participatory governance by Escobar, O.
- The epistemic basis of democratic deliberation by Mansbridge, J., Dryzek, J. S., Niemeyer, S., & Warren, M. E.

---

### Decision · Program_Chairs · 2025-10-08

**Decision:**

Reject

**Comment:**

Thank you for submitting to Agents4Science 2025! We regret to inform you that your submission has not been accepted. Please see the reviews below for more information.